# THE TAIL TELLS ALL: ESTIMATING MODEL-LEVEL MEMBERSHIP INFERENCE VULNERABILITY WITHOUT REFERENCE MODELS

## ABSTRACT

Membership inference attacks (MIAs) have emerged as the standard tool for evaluating the privacy risks of AI models. However, state-of-the-art attacks require training numerous, often computationally expensive, reference models, limiting their practicality. We present a novel approach for estimating model-level vulnerability, the TPR at low FPR, to membership inference attacks without requiring reference models. Empirical analysis shows loss distributions to be asymmetric and heavy-tailed and suggests that most points at risk from MIAs have moved from the tail (high-loss region) to the head (low-loss region) of the distribution after training. We leverage this insight to propose a method to estimate model-level vulnerability from the training and testing distribution alone: using the absence of outliers from the high-loss region as a predictor of the risk. We evaluate our method, the TNR of a simple loss attack, across a wide range of architectures and datasets and show it to accurately estimate model-level vulnerability to the SOTA MIA attack (LiRA). We also show our method to outperform both low-cost (few reference models) attacks such as RMIA and other measures of distribution difference. We finally evaluate the use of non-linear functions to evaluate risk and show the approach to be promising to evaluate the risk in large-language models.

## 1 INTRODUCTION

Large-scale machine learning systems have seen rapid adoption across healthcare, finance, legal, and other domains in recent years, with many models now being fine-tuned on sensitive and confidential data. A large body of research has however shown that these models may inadvertently memorize certain samples within their training data (often outliers) (Carlini et al., 2019; 2022b), raising strong privacy concerns.

Membership inference attacks (MIAs) have become the primary tool to measure the empirical privacy leakage of a model, evaluating the practical risk and providing a bound on the success of other types of attacks (Ponomareva et al., 2023). The privacy risk of a model is quantified by the attack's True Positive Rate (TPR) at a low False Positive Rate (FPR) (Carlini et al., 2022a; Watson et al., 2021; Ye et al., 2022; Zarifzadeh et al., 2023), focusing on the members an attack can confidently infer. This is also aligned with legal requirements, such as EU GDPR "reasonably likely" standard for singling out and the UK Information Commissioner's Office's recent guidance (European Data Protection Board, 2024; , ICO).

### 1.1 PROBLEM STATEMENT

The most effective MIAs, such as the Likelihood Ratio Attack (LiRA), however require training tens to hundreds of reference models, each requiring computational resources equivalent to the original model. The low loss of a sample on its own can be caused by both generalization and memorization. State-of-the-art MIAs are believed to mainly use reference models to distinguish between the two. It has also been shown that the success of an attack increases with the number of reference models, with recent work showing continuous improvement with 256 reference GPT-2 models (Hayes et al., 2025), and that reference models need to be trained with the same architecture and setup as the model under attack to be optimal (Carlini et al., 2022a; Zarifzadeh et al., 2023). The high computational

cost of SOTA attacks limits their practicality as privacy risk evaluation tools, especially for complex models and LLMs. Importantly, this cost becomes even more prohibitive in realistic workflows such as hyperparameter and architecture selection, where practitioners must compare privacy-utility tradeoffs across many training configurations Ponomareva et al. (2023), as well as continuous monitoring of routinely retrained production models and rapid pre-deployment risk assessments. All of these settings require evaluating numerous candidate models without incurring the computational overhead of training new reference models for every iteration.

A recent line of research has aimed to decrease the number of reference models needed to perform attacks. The state-of-the-art for this, RMIA, however still requires the training of at least one reference model (Zarifzadeh et al., 2023). Another line of research focused instead on methods to score the vulnerability of individual training samples to MIAs. These however do not provide an overall model-level risk metric (Leemann et al., 2024; Pollock et al., 2025).

**We here propose a new approach and method to estimate the vulnerability of a model to state-of-the-art membership inference attacks, leveraging new insights and using metrics available with no additional training.**

### 1.2 OUR APPROACH: MEASURING WHAT'S MISSING

Membership inference attacks take advantage of the fact that models tend to be more confident and exhibit lower loss in their predictions for data samples they have seen during training. This differential behavior is a direct consequence of the optimization process that aims to minimize the loss with some samples being more difficult to learn than others due to their difficulty, rarity or position as outliers in the dataset distribution and being memorized.

We study train and test loss distributions across a wide range of setups and observe empirically that loss distributions tend to be asymmetric and heavy-tailed, with most samples having a near zero loss whilst a few have a much higher loss than average. We posit that the heavy tail of the test set's distribution, compared to that of the train set, is due to samples from the training set's distribution that should have a high loss (hard example) but have shifted to the low-loss region during training, having been effectively memorized.

These observations are in line with the literature and explain a well-known fact: the LOSS attack (Yeom et al., 2018) often achieves poor TPR@low FPR values despite a relatively high AUC.

We also posit and empirically confirm across a wide variety of setups that samples missing from the tail of the training set distribution (high loss) constitute a significant fraction of the samples ultimately deemed to be most vulnerable to SOTA reference model-based MIAs.

Leveraging these insights, we propose a class of methods to estimate the model-level risk to SOTA MIAs from readily available train and test set loss distributions, measuring the absence of samples from the train distribution to estimate risks. In particular we show the TNR of the loss attack, its known ability to confidently identify non-members, to provide an accurate estimate of model-level risk.

## 2 PRELIMINARIES

### 2.1 NOTATION

We denote with $f_\theta : \mathcal{X} \to \mathcal{Y}$ a machine learning classification model parameterized by $\theta$, with input space $\mathcal{X}$ and output space $\mathcal{Y} = [0, 1]^n$. Model $f_\theta \leftarrow \mathcal{T}(D_{\text{train}})$ is trained on dataset $D_{\text{train}} \sim \mathcal{X} \times \mathcal{Y}$ using algorithm $\mathcal{T}$ and loss function $l$. A separate test dataset $D_{\text{test}} \sim \mathcal{X} \times \mathcal{Y}$ is reserved for evaluation and is never used during training. For sample $(x, y)$, we write $f_\theta(x)_y$ for the predicted probability for class $y$ and $\ell(x, y) = \ell(f_\theta(x)_y, y)$ for the corresponding loss.

Membership inference attacks (MIAs) aim to determine whether a given record $(x, y)$ was used to train target model $f_\theta$. Specifically, given target model $f_\theta$ and target record $(x, y) \in \mathcal{X} \times \mathcal{Y}$, an MIA $A(x, y, f_\theta)$ aims to infer whether $(x, y) \in D_{\text{train}}$ ($(x, y)$ is a *member*) or not ($(x, y)$ is a *non-member*). We denote the target model's loss on $(x, y)$ with $\ell_{\text{target}}(x, y)$. The attack outputs a membership score

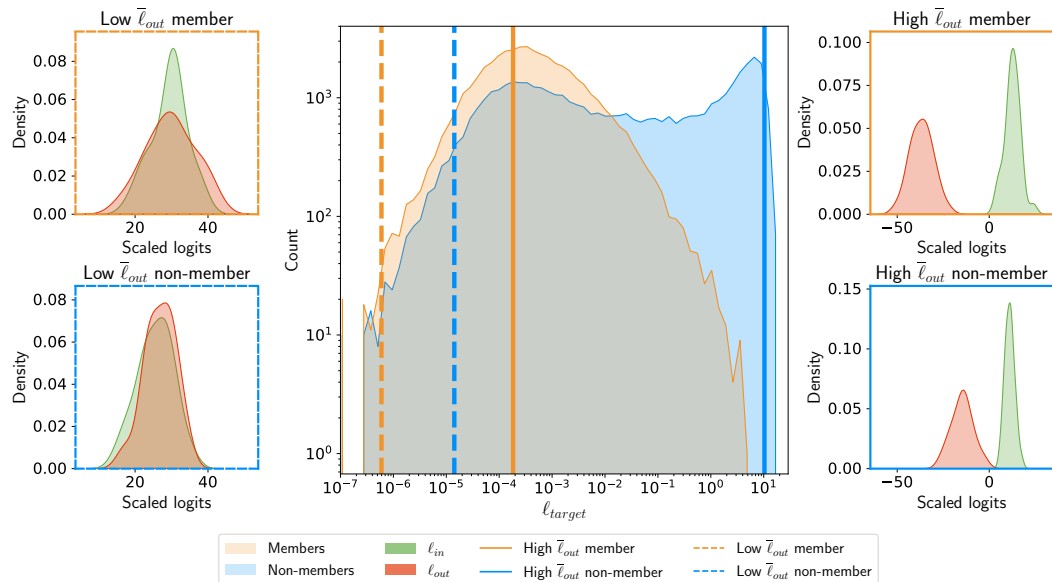

Figure 1: **Vulnerable members are mostly those that moved from tail to head.** The central panel shows member (orange) and non-member (blue) loss distributions on the target model. **Non-members dominate the high-loss tail** (right side), while members concentrate at low loss. Vertical lines mark $\ell_{target}$ values for representative sample groups. Side panels display the IN and OUT scaled logit distributions that LiRA uses to compute membership scores for these samples: left panels show samples with low $\ell_{out}$, right panels show samples with high $\ell_{out}$. In particular, note how the **LiRA-vulnerable members** (orange solid line, top-left panel) have low $\ell_{target}$ but high $\ell_{out}$ indicating these samples would naturally fall in the high-loss tail but moved to the low-loss $\ell_{target}$ head through training. Results shown are for ResNet-18 trained on CINIC-10.

that quantifies the likelihood of the target sample $(x, y)$ being part of the training data of the target model. The attacker then makes a binary membership prediction according to threshold $\tau$:

$$A_\tau(x, y, f_\theta) = \begin{cases} 1 & \text{if } A(x, y, f_\theta) \geq \tau \\ 0 & \text{otherwise} \end{cases} \tag{1}$$

A simple and widely used example is the LOSS attack $A_{\text{LOSS}}$ (Yeom et al., 2018), where a sample's membership score is the loss of the target model on it and $\mathbb{1}$ is the indicator function:

$$A_{\text{LOSS}}(x, y) = \mathbb{1}[-\ell_{\text{target}}(x, y) > \tau] \tag{2}$$

We report the model-level risk as the True Positive Rate (TPR) at a fixed False Positive Rate (FPR)=$\alpha$ (Carlini et al., 2022a; Ye et al., 2022; Zarifzadeh et al., 2023) of an attack.

SOTA attacks rely on *reference models*, models trained on auxiliary datasets $D_{\text{reference}} \sim \mathcal{X} \times \mathcal{Y}$ using the same architecture and training process as the target model. For a sample $(x, y)$, an "OUT" reference model $f_{\theta_{OUT}}$ is trained on data excluding $(x, y)$, while an "IN" reference model $f_{\theta_{IN}}$ is trained with it. This distinction enables the attacker to compare model behavior in the presence and absence of the target sample, improving its power. For a sample $(x, y)$, we denote with $\ell_{\text{OUT}}(x, y)$ the loss of an OUT model on $(x, y)$, and analogously for an IN model. For K IN reference models, we define the mean loss:

$$\bar{\ell}_{in}(x, y) = \frac{1}{K} \sum_{k=1}^{K} \ell(f_{\text{IN}}^k(x, y)) \tag{3}$$

and analogously for $\bar{\ell}_{out}(x, y)$.

## 2.2 SOTA REFERENCE MODEL-BASED MEMBERSHIP INFERENCE ATTACKS

The current state-of-the-art reference-model based attack is the Likelihood Ratio Attack (LiRA) (Carlini et al., 2022a). It estimates the loss distributions of models trained with and without a target sample $(x, y)$, and computes their likelihood ratio as a membership score. In the *online* attack, the attacker trains both "IN" and "OUT" reference models, and in the more efficient *offline* version, only trains "OUT" models. We adopt the strongest online attack in our experiments.

Recent work has explored methods to reduce the number of reference models needed for an effective attack. The state-of-the-art approach in this setting is RMIA (Zarifzadeh et al., 2023). It computes membership scores by comparing the target sample to samples sampled from the population using simple pairwise tests, and has been show to outperforms LiRA when few reference models are available. We compare our method to it.

## 3 EXPERIMENTAL SETUP

**Datasets.** We select 4 image classification datasets widely used in the literature: MNIST, CIFAR-10, CINIC-10 and CIFAR-100. MNIST contains 70,000 grayscale 28×28 images of handwritten digits (0-9). CIFAR-10 has 60,000 32×32 color images across 10 natural object classes, and CINIC-10 combines CIFAR-10 with downsampled ImageNet data, creating 270,000 32×32 images across the same 10 classes as CIFAR-10 but with greater data volume and diversity. Finally, CIFAR-100 contains the same images as CIFAR10, but is divided into 100 fine-grained classes.

**Models.** We train 9 neural network architectures commonly used for image classification, ranging between 60K and 172M parameters on all 4 datasets: ResNet-20 (He et al., 2016), WRN28-2 (Zagoruyko & Komodakis, 2016), MobileNetV2 (Sandler et al., 2018), DenseNet121 (Huang et al., 2017), WRN40-4, ResNet-18, WRN28-10, VGG11 and VGG16 (Simonyan & Zisserman, 2015). Additionally, we finetune ViT-B-16 pretrained on ImageNet on CIFAR-10. We follow the setup from Carlini et al. (2022a) where each models is trained on a randomly selected subset consisting of 50% of the samples from the training split as "members," with the other 50% reserved as non-members using regularization methods including weight decay and data augmentation to achieve a high training accuracy of over 90%. We use stochastic gradient descent as the optimizer with a cosine annealing learning rate schedule and a batch size of 256. Models are trained for 50 to 200 epochs depending on the complexity of the dataset and the size of the model.

**Attacks.** We instantiate LiRA and RMIA in the "online" setting with 64 reference models, comprising 32 IN and 32 OUT models. We follow the "mirror" setup from Carlini et al. (2022a) using 2 augmentations.

For the LOSS attack, we collect per-sample loss values at the final epoch of training for the target model.

## 4 INTUITION

**Loss distributions are heavy-tailed.** Figure 1 shows the member and non-member target model loss distributions on a semilog scale for ResNet-18 on CINIC-10; complete results are shown in Appendix A. In both cases, most probability mass lies at low loss (the "head"), with a smaller fraction at high loss (the "tail"). For members, the head corresponds to learned or memorized examples, while the tail comprises harder-to-learn samples (Pollock et al., 2025). For non-members, the head reflects samples the model generalizes to, whereas the tail contains difficult or outlier cases that the model fails to fit.

**The difference between member and non-member loss distributions is driven by the tail.** The member and non-member distributions differ in mean but overlap substantially in the low-loss region. The meaningful separation arises in the high-loss tail, where the non-member distribution places much more mass than the member distribution.

During the optimization process, gradient updates are likely larger for high-loss training examples; training pushes these extreme member losses into the low-loss region, shrinking the member tail rather than uniformly lowering all losses.

Because this tail is composed almost entirely of high-loss non-members, the LOSS attack can identify these samples with high confidence.

**Vulnerable members are those that are missing from the tail.** The non-member distribution reflects the model's behavior on unseen data. Assuming that members and non-members are drawn i.i.d. from the same underlying distribution, the absence of a heavy tail for members then suggests that samples which would be in the high-loss region, had they not been seen by the model during training, have shifted into the low-loss region. These members now fall into the head of the distribution, where they are difficult to distinguish from easy non-members.

State-of-the-art membership inference attacks such as LiRA target exactly these members. We therefore hypothesize that the members most vulnerable to LiRA are mainly these tail-to-head samples. Figure 1 supports this hypothesis: samples with the highest LiRA scores exhibit low target-model losses, yet the $\bar{\ell}_{out}$ is high and aligns with the heavy tail of the non-member distribution (see Appendix A for results for all setups). This pattern indicates that training shifted these examples from the tail to the head. We provide a formal explanation of this intuition in Appendix B.

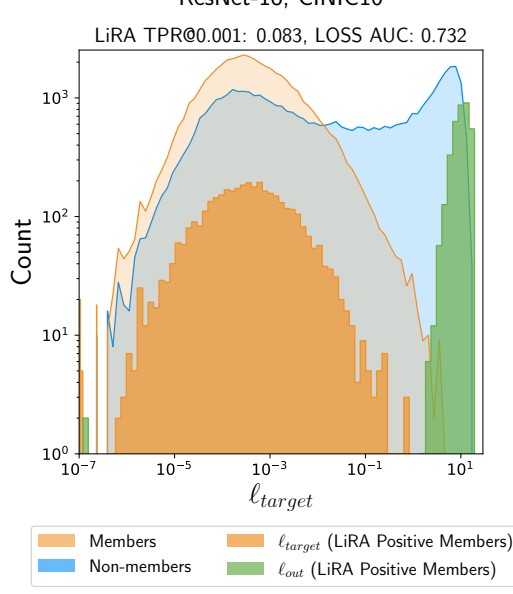

Figure 2: Loss distributions for members and non-members for ResNet-18 (CINIC10). Orange histogram shows member losses, blue shows non-member losses, with density curves overlaid. Green bars indicate the OUT model mean ($\mu_{out}$) for points identified by LiRA at FPR=0.001.

## 5    ESTIMATING VULNERABILITY TO SOTA MIAS

Leveraging these insights, we propose a new approach and method to evaluate model-level risk to SOTA MIAs that does not require expensive reference model training: measuring the "missing" member records. In particular, we propose to estimate model-level risk using the True Negative Rate (TNR) at a fixed FPR of the LOSS attack. The TNR measures how good the LOSS attack is at confidently identifying non-members, providing a measure of how "separable" the tail is from the member distribution. For a set of non-members $D_{\text{test}}$, model $f_\theta$ and threshold $\tau$, we compute the LOSS TNR as the fraction of correctly identified non-members:

$$\text{TNR}_{\text{LOSS}}(D_{\text{test}}, f_\theta, \tau) = \frac{|\{A_{\text{LOSS}}(f_\theta, x, y) > \tau | (x, y) \in D_{\text{test}}\}|}{|D_{\text{test}}|} \tag{4}$$

We select $\tau$ to achieve a False Negative Rate equal to the FPR of the LiRA attack for which we are estimating the TPR. In line with the literature, we focus on FPR=0.001, but present results for different FPR values (see Appendix B). We select FNR=FPR as the most natural choice, but show our method is robust to the choice of FNR threshold: as shown in Appendix B, LOSS TNR maintains strong predictive performance across a wide range of FNR values, with RMSE remaining relatively stable.

Figure 3 shows the LOSS TNR to be a very good predictor of LiRA's TPR@FPR=$10^{-3}$ across models and datasets, obtaining a low RMSE of 0.03 with a simple linear predictor with one parameter (we fix the y-intercept to be 0 as we expect a LOSS TNR of 0 to result in a LiRA TPR of 0). Bootstrapping with 1000 iterations provides narrow confidence intervals, confirming that the linear

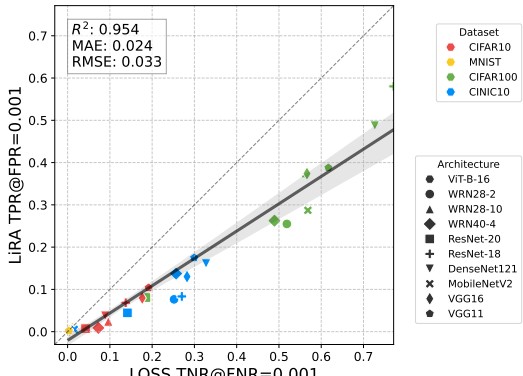

Figure 3: LOSS TNR reliably predicts LiRA TPR@FPR=0.001 across varied model architectures and datasets. The linear relationship (solid black line) exhibits strong performance with RMSE=0.033. The 97.5% confidence interval is shown as the shaded region.

Table 1: Performance metrics for LOSS TNR@$0.001$ as a predictor of LiRA TPR@$0.001$. Results are averaged across datasets and architectures reported in Section 3.

| Metric | $R^2$ | RMSE | MAE |
|---|---|---|---|
| LOSS TNR (Ours) | **0.954** | **0.032** | **0.024** |
| Train-Test Gap | 0.717 | 0.081 | 0.060 |
| LT-IQR AUC | 0.806 | 0.067 | 0.046 |
| Loss AUC | 0.844 | 0.051 | 0.052 |
| RMIA (2 reference models) | 0.910 | 0.046 | 0.036 |

model maintains a high degree of certainty over the full range of values. This allows practitioners to accurately estimate the LiRA TPR@FPR without reference models. This is particularly useful for many scenarios that require evaluating many candidate models such as hyperparameter tuning, neural architecture search, or comparing training strategies.

We then compare our approach against an alternate metric, (i) LOSS attack AUC, and baselines: (ii) train-test accuracy gap of the target model; and (iii) LT-IQR AUC (Pollock et al., 2025). LT-IQR is the SOTA for identifying the samples most at risk to LiRA, but is not a measure of overall model vulnerability.

Table 1 reports goodness-of-fit metrics for linear functions fit on our approach and baselines and show it to outperform the baselines and to be a good estimator of vulnerability to LiRA.

Aware of the prohibitive cost of SOTA MIA methods, a recent line of research aims to develop low-costs attacks which can be used by model developers to estimate the model-level risk. We also instantiate the SOTA low-cost attack, online RMIA, with two reference models and compare its predictive power to our method.

As shown in Table 1, our method outperforms RMIA at estimating LiRA TPR, despite RMIA requiring at least 2 reference models.

LOSS AUC, which measures distributional separability across all loss values, would be more effective if memorization were evenly distributed. The train-test gap, a metric studied in the literature, captures only mean differences between loss distributions. While it performs moderately well, we observe that it cannot detect tail differences caused by the asymmetric non-member distribution.

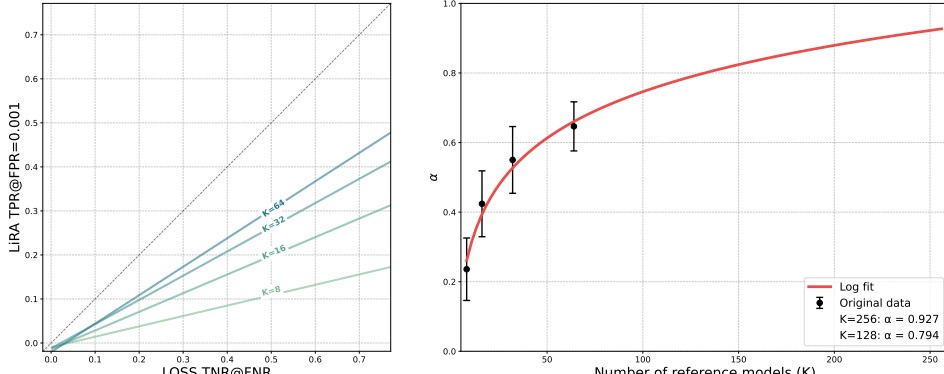

Figure 4: Performance of linear models for predicting LiRA with varying numbers of reference models (K). Left panel shows the correlation between LOSS TNR (x-axis) and LiRA TPR@0.001 (y-axis) across different K values, with linear regression fits shown for each condition. Right panel shows slope parameter ($\alpha$) for different K. The error bars show the 97.5% confidence intervals obtained with boostrapping.

## 5.1 VARYING THE NUMBER OF REFERENCE MODELS

We study the effectiveness of LOSS TNR at estimating the TPR@FPR of attacks of varying strength. We instantiate LiRA with 4, 8, 16, 32 and 64 reference models and report the resulting linear model for each value of K, along with the slopes $\alpha$.

Figure 4 shows the risk to steadily increase with K as the attack becomes more confident at identifying members. As noticed in previous work though, TPR@low FPR experiences decreasing returns as the number of reference models increase which we can now model to estimate the risk against stronger attackers.

## 5.2 FITTING DIFFERENT FUNCTIONS

We have so far estimated LiRA TPR with a simple linear model. We now study whether non-linear models lead to better risk estimates. Identifying non-members is indeed a matter of finding a good threshold to separate the tail of the loss distribution from the rest, while identifying members is dependent on reference models and is likely to be a more difficult task. We indeed find empirically that the LOSS TNR is typically higher than LiRA TPR at the same FPR.

We thus compute goodness-of-fit metrics for convex functions such as a two-parameter polynomial, power-law, and exponential functions.

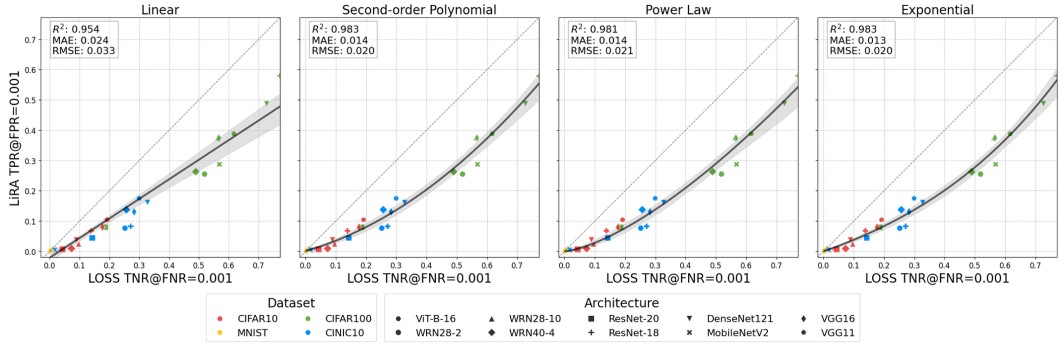

Figure 5: Comparison of regression models for predicting TPR@0.001 with LOSS TNR@0.001 evaluated on image recognition setups. The exponential function achieves the lowest RMSE of 0.02, compared to 0.033 achieved by the linear fit.

Figure 5 shows that the exponential function $a(e^{bx} - 1)$ achieves the best fit, outperforming the linear model. An exponential fit would imply that as LOSS TNR increases, member identification becomes easier as the model memorizes more difficult samples. Initial gains in LOSS TNR produce modest improvements in LiRA TPR, but these improvements accelerate as memorization increases.

### 5.3 ROBUSTNESS ACROSS LEVELS OF GENERALIZATION GAPS

Prior work has identified overfitting, typically measured by the train-test accuracy gap, as a key factor influencing membership inference vulnerability (Carlini et al., 2022a; Yeom et al., 2018).

We now study the effectiveness of our method for predicting the vulnerability of models with widely varying train-test gaps, ranging from near-zero (minimal overfitting) to over 50 percentage points (severe overfitting). Specifically, we treat intermediate training checkpoints from each setup in our main experiment as additional target models, instantiate LiRA against each target, and assess how well our method predicts its vulnerability. Figure 6 shows that LOSS TNR remains a reliable predictor of model vulnerability across this full range of generalization gaps.

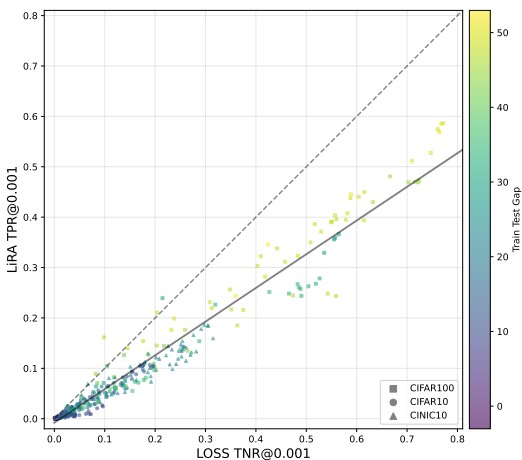

Figure 6: LOSS TNR reliably predicts LiRA TPR across varying overfitting levels.

This consistency demonstrates that LOSS TNR captures memorization patterns much richer than that of the overall generalization performance. While models with larger train-test gaps tend to have higher vulnerability (evidenced by the color gradient), TNR provides accurate risk estimates across a wide spectrum.

### 5.4 LLMS

Recent evidence suggests that LLMs exhibit fundamentally different memorization patterns than traditional models. The size of LLMs, for instance, allows them to achieve excellent generalization while simultaneously memorizing training data (Tirumala et al., 2022). The nature of their training data also makes them subject to what Shilov et al. (2024) describe as "mosaic memory", the ability of LLMs to memorize fuzzy duplicates in their training data, rather than exact samples.

Given the importance of the task, however, we test our approach on LLMs. Following the setup proposed in recent work Hayes et al. (2025), we evaluate our method on 5 GPT-2 models with 10M, 104M, 302M, 604M, and 1018M parameters. We train each model on a $2^{19}$-sample subset of the C4 dataset and instantiate online LiRA with 256 reference models (128 IN, 128 OUT).

While more traditional image and other models show discrete, example-specific memorization that creates clear distributional differences and asymmetric, heavy-tailed distributions, LLMs show loss distributions that are much more symmetrical, with little difference between the train and test distributions even when models can be shown to have memorized training data. TNR is therefore not an appropriate risk estimator in this setting.

However, we posit that LOSS AUC, as a more general estimate of "missing records", may provide an accurate estimation of model-level risk. Fig.7 shows it to indeed be a good estimator of the risk, obtaining an RMSE of 0.01.

## 6 RELATED WORK

**Membership inference attacks.** MIAs aim to infer whether a data sample was used to train a given target model. Homer et al. (2008) originally introduced membership inference attacks (MIA)

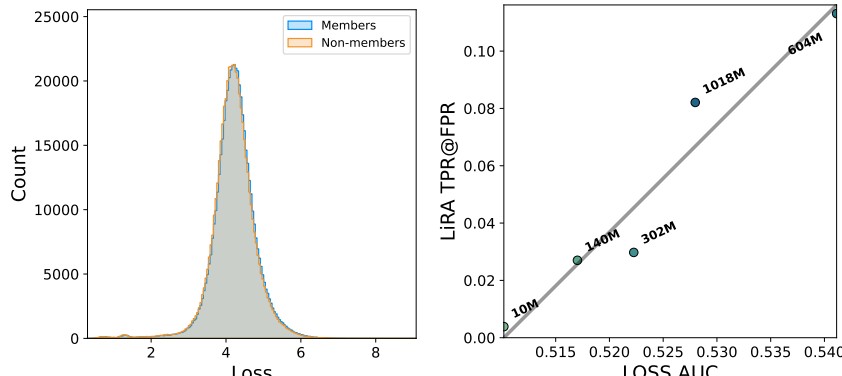

Figure 7: $\ell_{target}$ **distributions and predictive relationship for large language models**. Left: $\ell_{target}$ distributions for members (blue) and non-members (orange) show high overlap and symmetry, unlike the asymmetric, heavy-tailed distributions observed in image classification models (Figure 1). Right: Despite symmetric distributions, LOSS AUC reliably predicts LiRA TPR@FPR=0.001 across five GPT-2 models (10M to 1018M parameters), achieving RMSE=0.01. Points are labeled by model size.

to identify the presence of an individual's genome data among aggregate summary statistics. (Shokri et al., 2017) proposed the first MIA on machine learning models and demonstrated their efficacy in the black-box setting. MIAs have since become the de-facto standard to empirically measure the privacy risk posed by a model and providing an upper bound on the risk of more severe empirical attacks such as attribute inference or model inversion (Fredrikson et al., 2015).

Simple MIAs compare metrics like loss Yeom et al. (2018) or confidence Song et al. (2019) against thresholds, while others Mattern et al. (2023) compare samples against neighboring samples. While these attacks can obtain high AUC values, they have been shown to be unable to confidently identify members as they struggle to distinguish between low-loss samples that have been memorized and those that are easy to predict. Difficulty-calibrated attacks (He et al., 2024; Song & Mittal, 2020; Watson et al., 2021) aim to address this but still lack high confidence.

State-of-the-art attacks rely on reference models Shokri et al. (2017) to calibrate the attack with respect to sample difficulty. The Likelihood Ratio Attack (LiRA)(Carlini et al., 2022a) estimates loss distributions for IN and OUT models and performs a likelihood ratio test to determine membership of a given sample. Other methods such as Attack R use only OUT reference models (Ye et al., 2022), while others employ knowledge distillation to train reference models (Li et al., 2024; Liu et al., 2022).

**Low-cost attacks and free identification of vulnerable samples.** Reference model-based attacks depend heavily on the number and quality of reference models. On CIFAR-10, LiRA TPR@0.01% drops from $2.83\%$ with 254 reference models to $0.76\%$ with 4 (Zarifzadeh et al., 2023). Performance declines further with fewer models or smaller or mismatched architectures. In practice, dozens to hundreds of identical reference models are often needed for high accuracy, making these attacks difficult to scale to iterative workflows and large-scale models such as LLMs (Carlini et al., 2022a).

Recent work has explored ways to estimate model vulnerability to LiRA at lower computational cost or even for free. The state-of-the-art low-cost attack, RMIA (Zarifzadeh et al., 2023), combines reference models with population samples to construct an attack they show to match LiRA performance with as few as 2 reference models. Quantile regression-based attacks (Bertran et al., 2023) use a single regression model to match LiRA's TPR, but their effectiveness is limited to scenarios with only one reference model.

Another direction of research focuses on free identification of the samples that are most vulnerable to SOTA reference model-based MIAs. Predictors such as sample loss, confidence, and training loss dynamics have been shown to successfully identify highly vulnerable samples (Leemann et al., 2024; Pollock et al., 2025) for free. In particular, LT-IQR has been shown to achieve high precision

at identifying samples vulnerable to LiRA (Pollock et al., 2025). These methods, however, do not readily provide model-level risk scores as their outputs are inherently relative.

**Factors influencing membership inference vulnerability.** Previous work has explored the connection between a model's training and its vulnerability to MIAs. Yeom et al. (2018) proposed overfitting as a sufficient but not necessary condition for MIA success, whilst other works demonstrate a correlation between the train-test gap (a measure of overfitting) and MIA success (Carlini et al., 2022a; Yeom et al., 2018). Other work has studied the impact of dataset properties on model vulnerability: models trained on smaller datasets are more vulnerable (Chen et al., 2020; Németh et al., 2025; Tobaben et al., 2025), and, within a dataset, classes with fewer samples contain more vulnerable examples (Chang & Shokri, 2021; Kulynych et al., 2022).

## 7 CONCLUSION

We present a novel method to estimate model-level vulnerability to SOTA membership inference attacks without reference models. We show a significant fraction of highly vulnerable samples being samples that are "missing" from the tail of the training distribution and instead moved to the low loss head of the distribution. We propose to use the LOSS attack, and in particular its TNR, to quantify the missing samples and estimates the model's vulnerability to LiRA.

Across 9 architectures and 4 datasets, we show our method to achieves an RMSE of 0.032 in predicting the LiRA TPR@FPR=0.001 with similar results for other FPR values. Our method outperforms other measures including train-test accuracy gap and LT-IQR, as well as recent low-cost attacks at estimating model-level risk. We also test non-linear risk estimation models and show the method to enable model developer to estimate the risk posed by strong attackers willing to train a large number of reference models. Our method is robust to the generalization gap of the target model, and also to the choice of FNR threshold.

Finally, given the importance of the task, we apply our approach to LLMs. While they are know to exhibit different memorization pattern, we show LOSS AUC might still serve as an effective estimation of model-level vulnerability to SOTA reference model-based MIAs.

Taken together, our work provides a novel practical approach to estimating model-level vulnerability to SOTA attacks with no additional computational cost. Where SOTA attacks often require training hundreds of reference models, our method uses only the target model's loss values. We believe this will greatly help enable privacy risk assessments in practice, in particular during iterative development workflows and scenarios where multiple candidate models require assessment.

**Limitations and future work.** We evaluate in the work the use of LOSS TNR (and AUC) to estimate model vulnerability to LiRA. While LiRA is the state-of-the-art and has been shown to outperform other approaches, we have not assessed the transferability of our method to other MIAs or to broader privacy threats (e.g., reconstruction or inversion). While we test it on a range of setups (datasets and architectures) covering a range of risk level, TNR is an estimator of the risk and there might be cases where our results do not generalize. Due to computational constraints, our LLM study spans five mid-sized architectures. Evaluating scaling to larger models and datasets and sensitivity to training regimes are natural next steps.

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

# 8 APPENDIX

## A LOSS DISTRIBUTIONS ACROSS ALL SETUPS

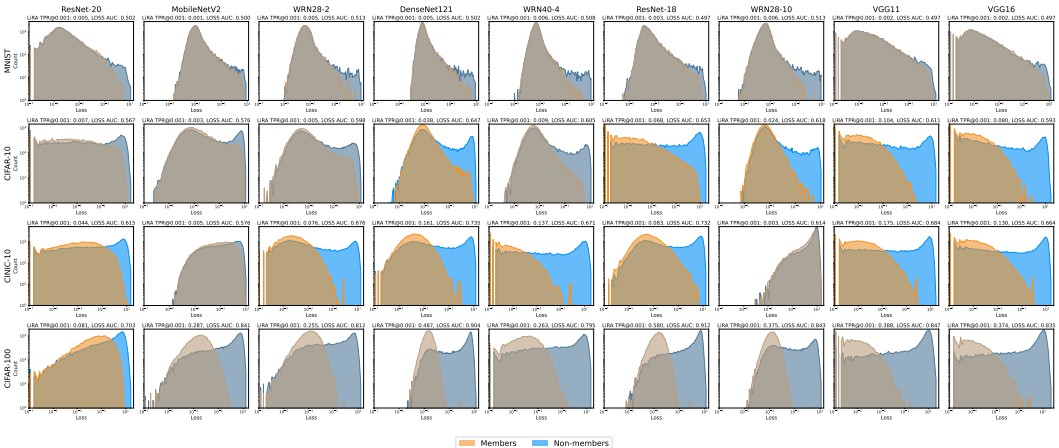

Figure 8: Histograms showing $\ell_{target}$ distributions for training set members (orange) and non-members (blue) in log-log scale across all setups. The distributions demonstrate clear separation between members and non-members, with members typically having lower loss values.

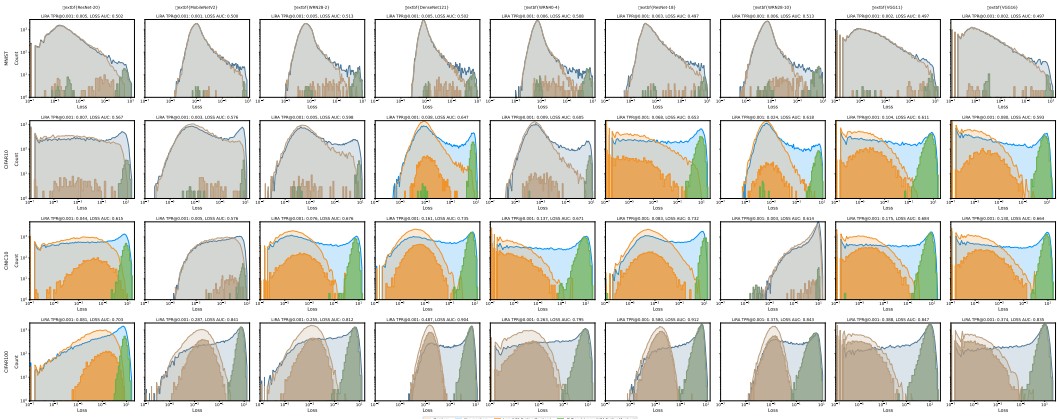

Figure 9: $\ell_{target}$ distributions for training set members and non-members across all setups. Orange histogram shows member losses, blue shows non-member losses, with density curves overlaid. Green bars indicate the OUT model mean ($\ell_{out}$) for points identified by LiRA at FPR=0.001.

# B VARYING FPRs

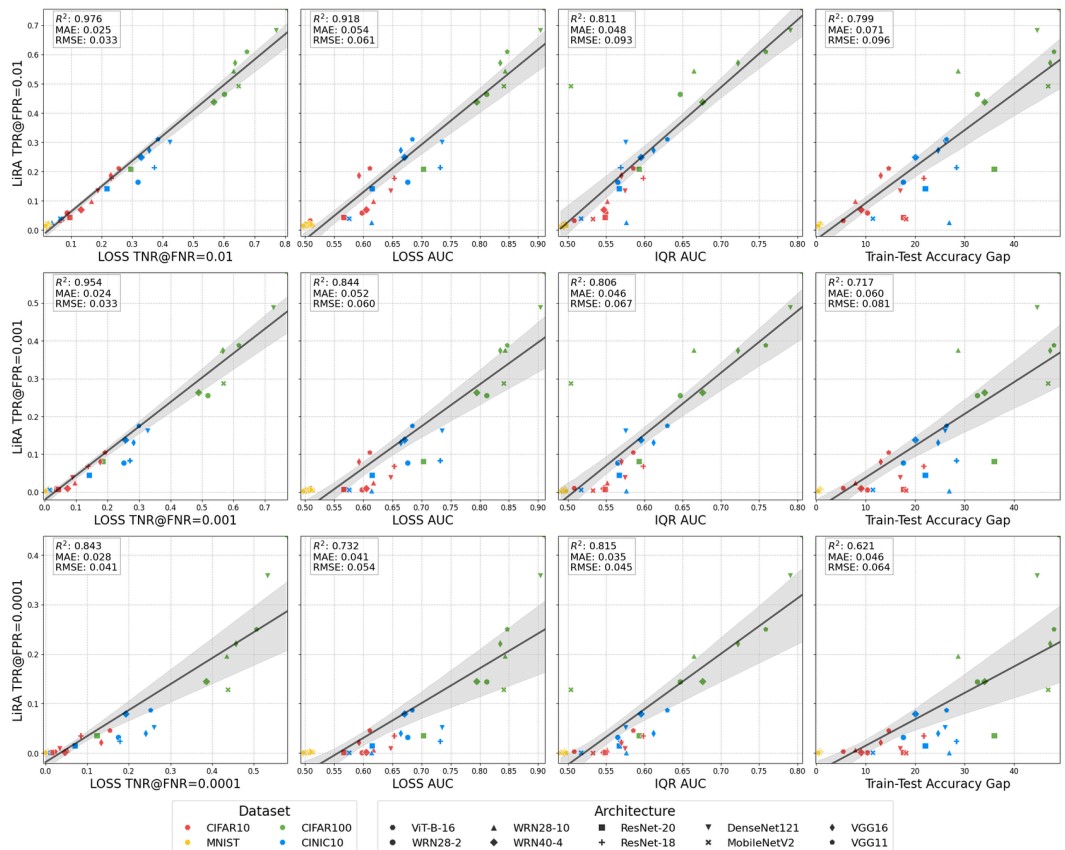

Figure 10: Comparison of different metrics for predicting LiRA TPR at FPRs of 0.01, 0.001, 0.0001. Each panel shows the relationship between a predictor metric (x-axis) and LiRA TPR (y-axis) across all model-dataset combinations. Linear fits (solid lines) with 97.5% confidence intervals (shaded regions) show goodness-of-fit.

# C SENSITIVITY TO SELECTION OF LOSS FNR THRESHOLD

We select FPR=FNR across the paper, as it is a natural choice. We here study the sensitivity of LOSS TNR's predictive capability to the choice of FNR threshold. Following the experimental setup described in Section 3, we evaluate how well LOSS TNR predicts LiRA's TPR across four different false positive rate thresholds: 0.1, 0.01, 0.001, and 0.0001. For each LiRA FPR setting, we fit a linear function to predict the corresponding LiRA TPR from LOSS TNR values computed at varying FNR thresholds, and measure prediction quality using the RMSE.

Figure 11 shows that RMSE remains relatively stable across a broad range of FNR values, demonstrating that LOSS TNR's predictive power is robust to the specific threshold selection. Notably, setting the FNR equal to the target LiRA FPR (green stars) achieves a similar performance compared to the FNR that minimizes RMSE (yellow stars), showing it to be a good and principled choice. The sharp degradation in RMSE only occurs at very high FNR values (approaching 1.0), where the LOSS threshold becomes too relaxed to provide a meaningful signal.

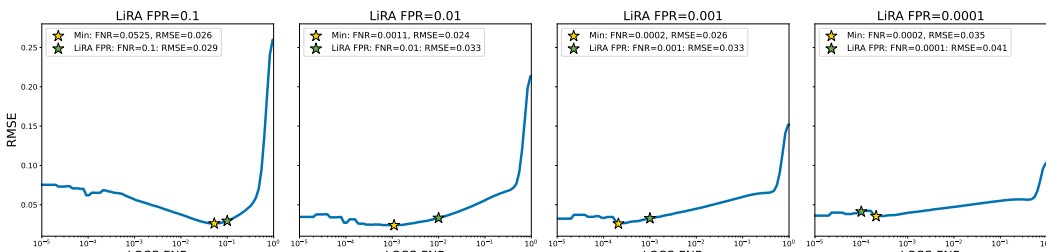

Figure 11: RMSE of LOSS TNR@different FNR values as a predictor of LiRA TPR at a fixed FPR. The yellow star indicates the FNR that achieves the lowest RMSE, while the green star shows performance when setting the FNR to equal to the LiRA FPR. Across all settings, the RMSE difference between the optimal selection and FNR=FPR is minimal.

## D   EXTENDED INTUITION

We provide a formal explanation of the connection between LiRA TPR@FPR and the LOSS attack TNR@FNR. For a record $(x, y)$, we use the notation from Section 4 for the mean IN and OUT losses, $\ell_{\text{in}}$ and $\ell_{\text{out}}$.

Consider a target model $f_\theta \leftarrow \mathcal{T}(D_{\text{train}})$. We refer to records contained in $D_{\text{train}}$, i.e., records the model has seen during training, as *members*, and to records in $D_{\text{test}}$, i.e., records the model has not seen during training, as *non-members*.

For a record $(x, y)$, we denote $Q_{\text{in}}$ and $Q_{\text{out}}$ the distributions of its IN and OUT losses, respectively, and denote its LiRA score by $\Lambda(x, y)$.

For a member $(x, y)$, if $Q_{\text{in}}$ and $Q_{\text{out}}$ differ substantially, training the model on $(x, y)$ has a large effect on its loss on $(x, y)$: the loss typically moves from high values (in the tail of the loss distribution) to low values (in the head). In this case, $\Lambda(x, y)$ is likely to be high, making $(x, y)$ highly vulnerable to LiRA. We call such records *tail-to-head* records and denote the set of all such training records by $D_{\text{th}}$.

At some LiRA threshold $\tau_{\text{LiRA}}$, $D_{\text{th}}$ makes up a fraction $\alpha$ of the highly vulnerable records in $D_{\text{train}}$:

$$\left| D_{\text{th}} \right| = \alpha \left| \left\{ (x, y) \in D_{\text{train}} : \Lambda(x, y) \geq \tau_{\text{LiRA}} \right\} \right|, \tag{5}$$

$$\frac{\left| D_{\text{th}} \right|}{\left| D_{\text{train}} \right|} = \alpha \frac{\left| \left\{ (x, y) \in D_{\text{train}} : \Lambda(x, y) \geq \tau_{\text{LiRA}} \right\} \right|}{\left| D_{\text{train}} \right|}. \tag{6}$$

The quantity on the right-hand side of the second line is precisely the LiRA true positive rate at threshold $\tau_{\text{LiRA}}$, i.e.,

$$\frac{\left| D_{\text{th}} \right|}{\left| D_{\text{train}} \right|} = \alpha \, \text{TPR}_{\text{LiRA}}(\tau_{\text{LiRA}}). \tag{7}$$

Thus, the LiRA TPR is proportional to the fraction of tail-to-head records in $D_{\text{train}}$, and this fraction characterizes model-level vulnerability.

We now consider the records in $D_{\text{test}}$. Records in $D_{\text{test}}$ that lie in the tail of the loss distribution under $f_\theta$ are likely to have high OUT loss, i.e., large loss when they are not seen during training. If we were to train on such a record, its loss would either (i) remain in the tail (the model does not learn it) or (ii) move to the head (the model memorizes it). Empirically, training loss distributions are typically not heavy-tailed. We therefore assume that most records in $D_{\text{test}}$ that are in the tail of the OUT-loss distribution would move to the head if they were included in training. These are the analogous to the tail-to-head records in $D_{\text{train}}$.

Assuming that $D_{\text{train}}$ and $D_{\text{test}}$ are drawn i.i.d. from the same underlying distribution, we can estimate the fraction of tail-to-head records in $D_{\text{train}}$ by the corresponding fraction of tail records in $D_{\text{test}}$ at

some LOSS threshold $\tau_{\text{LOSS}}$:

$$\frac{\left|\{(x,y) \in D_{\text{test}} : \ell_{\text{out}}(x,y) \geq \tau_{\text{LOSS}}\}\right|}{|D_{\text{test}}|} = \frac{|D_{\text{th}}|}{|D_{\text{train}}|} + \varepsilon, \tag{8}$$

for some small error term $\varepsilon$.

The left-hand side is exactly the TNR of the LOSS attack at threshold $\tau_{\text{LOSS}}$. Hence,

$$\text{TNR}_{\text{LOSS}}(\tau_{\text{LOSS}}) = \alpha\, \text{TPR}_{\text{LiRA}}(\tau_{\text{LiRA}}) + \varepsilon. \tag{9}$$

This formalizes the intuition that the LOSS TNR provides a linear proxy for the LiRA TPR, up to a proportionality constant $\alpha$ and an error term $\varepsilon$.

# E  PREDICTING VULNERABILITY TO DIFFERENT ATTACKS

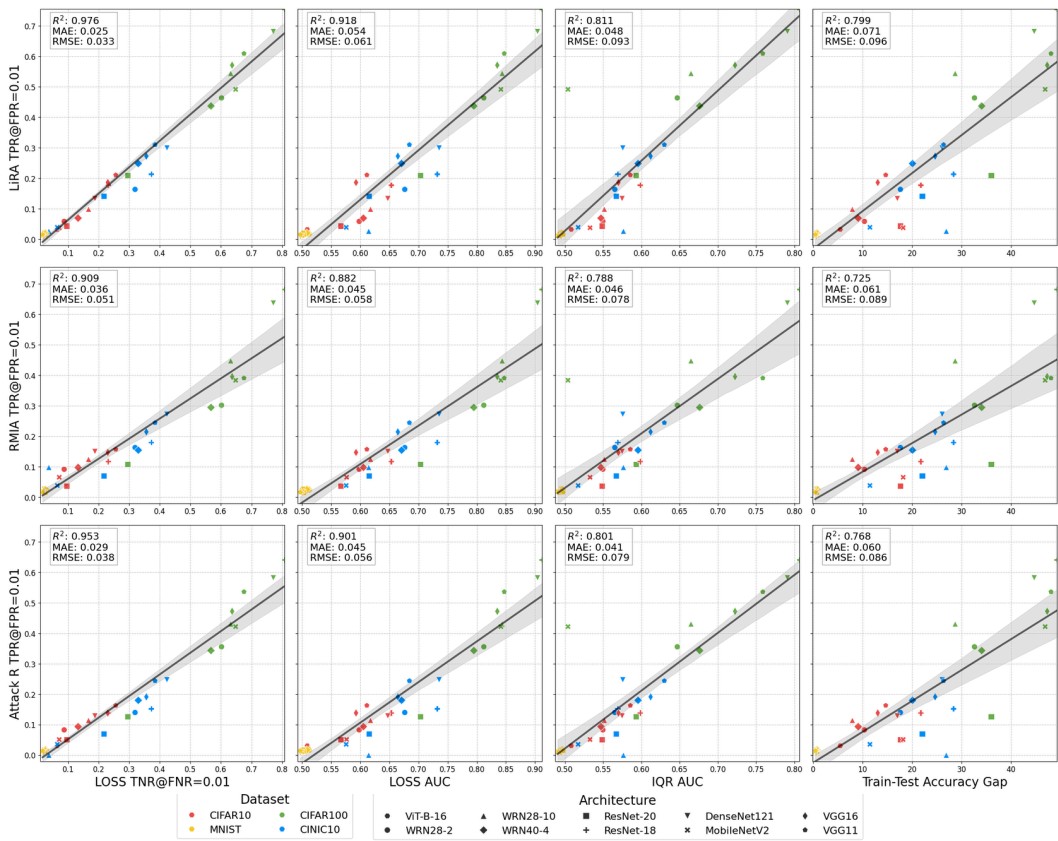

Figure 12: Comparison of different metrics for predicting the TPR at FPR=0.01 for LiRA, RMIA and Attack-R with 64 reference models. Each panel shows the relationship between a predictor metric (x-axis) and the attack TPR (y-axis) across all model-dataset combinations. Linear fits (solid lines) with 97.5% confidence intervals (shaded regions) show goodness-of-fit.

To evaluate whether our method generalizes to other attacks within the same family, we estimate the TPR@0.01 for RMIA and Attack R using 64 reference models (for RMIA: 32 IN and 32 OUT; for Attack R: 64 OUT). Figure N demonstrates that TNR serves as an effective predictor for both RMIA and Attack R (two reference-model-based MIAs). Since these attacks share a common approach of using reference models to estimate counterfactuals for individual samples, it follows logically that TNR captures the same underlying signal across attacks in this family.

