# OpenReview forum: "The Tail Tells All: Estimating Model-Level Membership Inference Vulnerability Without Reference Models"
_ICLR.cc/2026/Conference — Submitted to ICLR 2026_

### Official Review · Reviewer_7a13 · 2025-10-26

**Soundness:** 3
**Presentation:** 1
**Contribution:** 2
**Rating:** 4
**Confidence:** 4

**Summary:**

The authors propose a new metric to approximate the vulnerability of a model to SOTA MIAs without the need to train additional reference models. As empirical analysis shows, loss distribution on training data points tends to be long-tailed. Their proposed metric is based on the idea that during training, atypical training points tend to move from high-loss (tail) to low-loss (head) of the loss distribution as a consequence of memorisation. Identifying such samples can then provide an estimate of MIA vulnerability. The proposed metric estimates the true negative rate (TNR) of a model at a fixed FPR for an attack to account for such tail-to-head records.

**Strengths:**

- The proposed approach does not rely on reference models and is computationally feasible.
- The new metric is tested against a diverse set of models (including LLMs) and image classification datasets.
- The estimated MIA vulnerability using the new metric is compared against MIA vulnerability estimates using other SOTA metrics in the literature, such as LT-IQR.
- Authors(s) use the proposed metric to estimate MIA vulnerability for SOTA MIAs such as LiRA and RMIA (with 64 shadow models), thereby providing empirical evidence supporting the validity of their method.

**Weaknesses:**

- The metric appears to be sensitive to the choice of pre-set threshold used to compute LOSS TNR at fixed FPR as demonstrated by Eq (4). It is possible that the choice of threshold changes depending on the choice of datasets/ models. Furthermore, it is unclear whether the author(s) vary the threshold for different experimental settings. Nor is it clear if their proposed metric is robust to the choice of threshold.
- In Line 344, the authors conjecture "An exponential fit would imply that as LOSS TNR increases, member identification becomes easier as the model memorises more difficult samples...". But they provide no empirical evidence to support this. As a reviewer, I would appreciate a plot (if feasible) equivalent of Table 3. Furthermore, it is not specified in the paper if the results shown in Table 3 pertain to a specific training setting or it is generalizable to other datasets/models.
- The figures/ tables in the paper lack the necessary information to clearly convey the author(s)' intended message. They are supposed to be self-contained with as little need to refer to text in the paper as possible:
    - Table 1: Does "Ours" in the table refer to the Loss TNR at fixed FPR? Can you make it clear in the caption or the table?
    - Table 2: Same as Table 1. It lacks the context to interpret the contents of the table.
    - Table 3: Same as Table 1. It lacks the context to interpret the contents of the table.
- In all figures, you use TNR@FNR instead of TNR@FPR.
- Line 335: I suppose you mean “that a linear model may not be the most appropriate given the task at hand.”
- Figure 5's caption says, "The LIRA TPR@FPR LOSS as a linear function of the LOSS AUC evaluated on LLMs." Assuming this refers to the subfigure on the right, the metric in the subfigure on the x-axis is LOSS TNR@FNR and not the LOSS AUC as the caption suggests. It will cause confusion about the effect/ claim the author(s) intend to convey using the figure.

**Questions:**

**Questions**: If the author(s) can address the weaknesses detailed above, I would be amenable to revising my initial assessment.

**Suggestions**:
The author(s) provide ample evidence (in Section 4, Hypothesis) to support their proposed metric, but there is a lack of empirical evidence that would be necessary to solidify their argument, as detailed among the weaknesses. A bigger issue with the paper is its poor presentation. The author(s) need to improve their presentation, which, in its current version, makes the paper a rather difficult read. I have detailed some of the presentation issues in the weaknesses.

---

> ### Author Response · Authors · 2025-11-24
>
> We thank the reviewer for their thorough review and detailed feedback on presentation issues. We appreciate the recognition of our method's computational feasibility, diverse experimental evaluation, and comparison against SOTA metrics. We address each concern below.
>
> >“The metric appears to be sensitive to the choice of pre-set threshold used to compute LOSS TNR at fixed FPR as demonstrated by Eq (4). It is possible that the choice of threshold changes depending on the choice of datasets/ models. Furthermore, it is unclear whether the author(s) vary the threshold for different experimental settings. Nor is it clear if their proposed metric is robust to the choice of threshold.”
>
> Thank you for raising this point. We apologize that our threshold selection methodology was not sufficiently clear. **We have now added a threshold sensitivity analysis to Appendix C, and show that our method remains effective at a wide range of low LOSS FNR values.**
>
> We originally set the LOSS threshold so that the LOSS FNR matches the FPR at which we evaluate LiRA. For example, we predict LiRA TPR@FPR=10-2 with LOSS TNR@FNR=10-2, i.e. the fraction of non-members correctly identified by LOSS attack when it misclassifies 1% of the members as non-members.
>
> We, however, agree that while we view this as a natural and reasonable choice, further analysis of the sensitivity of the method to the threshold would provide meaningful insights. In a new additional experiment, we fix the LiRA FPR and vary the LOSS FNR, and report the results in Appendix C. The experiment shows LOSS TNR to remain an accurate predictor of LiRA TPR across a broad range of thresholds and validates that LOSS FNR = LiRA FPR is a reasonable choice.
>
> >In Line 344, the authors conjecture "An exponential fit would imply that as LOSS TNR increases, member identification becomes easier as the model memorises more difficult samples...". But they provide no empirical evidence to support this. As a reviewer, I would appreciate a plot (if feasible) equivalent of Table 3. Furthermore, it is not specified in the paper if the results shown in Table 3 pertain to a specific training setting or it is generalizable to other datasets/models.
>
> We thank the reviewer for this suggestion. **We now include plots equivalent to Table 3 in Section 5.2 of the revised manuscript.** Table 3 in the previous version showed results aggregated across all of our setups (excluding LLMs), as does Figure 3. We clarify this in the revised paper.
>
> >Regarding presentation issues:
>
> Thank you for the detailed feedback on figures and tables. We have addressed these points and polished the presentation of the entire paper in the revision.
>
> In summary, we:
> - Clarify the principled threshold selection methodology in the paper
> - Add analysis showing robustness to threshold choice across different FPR values
> - Add visualizations for Table 3
> - Improve all table captions to be self-contained with clear labeling
> - Correct all caption errors, and typos
>
> We hope these revisions address your concerns. Please let us know if there are any remaining issues, suggestions, or questions that -- if clarified or improved -- would contribute to raising the overall score.

---

> > ### Comment · Reviewer_7a13 · 2025-11-25
> > **Follow-up To The Rebuttal**
> >
> > I am satisfied with the efforts of the authors towards addressing my concerns and improving the paper. However, I do share the concern of other reviewers about the limited novelty of this work.

---

### Official Review · Reviewer_r89d · 2025-10-31

**Soundness:** 2
**Presentation:** 3
**Contribution:** 2
**Rating:** 2
**Confidence:** 4

**Summary:**

This paper proposes a method for estimating model-level vulnerability to membership inference attacks without training reference models. The key idea is that the absence of high-loss (tail) samples in the training loss distribution correlates with the model’s susceptibility to MIAs. The authors empirically show that the TNR of a simple loss-based attack can predict the TPR of LiRA at low false positive rates. The approach aims to offer a low-cost privacy risk estimation method suitable for large or resource-limited settings.

**Strengths:**

- The empirical results cover multiple datasets and architectures, demonstrating correlation between the proposed metric (LOSS TNR) and LiRA’s TPR at FPR.

- The paper is clearly written and easy to follow.

**Weaknesses:**

- The paper focuses on model-level vulnerability estimation, but it is unclear what the real-world application scenario of such a metric is. In privacy evaluation, MIAs are primarily defined as worst-case, sample-level privacy breaches (determining whether a particular record was in training), as evidenced by Carnili et al.. A model-level average metric offers little actionable guidance: practitioners either need to test specific data records (for auditing) or evaluate defense mechanisms under realistic attack settings. The proposed metric seems to produce a correlation measure with LiRA or RMIA, but it is unclear how this would be used in practice. The paper does not provide any concrete use cases or deployment scenarios.

- The technical novelty of the work is also limited. The core contribution amounts to computing the True Negative Rate (TNR) of a standard loss-based attack. This idea lacks theoretical grounding or statistical justification. The absence of analytical insights weakens the contribution, making it less suitable for a top-tier venue such as ICLR.

- The evaluation is restricted to LiRA and RMIA, both of which rely on output-distribution differences. However, many MIAs operate under different assumptions: Reference-calibrated or label-only attacks (e.g., He et al., 2024; Ye et al., 2022) rely on label confidence or query perturbations, not continuous loss values. It remains unclear whether the proposed estimator generalizes to those attack families.

- Finally, although the authors compare their approach with metrics like LT-IQR AUC and train-test gap, these baselines are not designed for model-level vulnerability estimation. As a result, the comparison does not convincingly demonstrate the proposed method’s superiority or distinct advantages.

**Questions:**

- What are the real-world application scenarios where a model-level vulnerability metric is practically useful for privacy evaluation?

- What theoretical or statistical justification supports using the TNR of a simple loss attack as a valid estimator of MIA vulnerability?

- Does the proposed method generalize beyond LiRA and RMIA to other attack types, such as label-only or reference-calibrated MIAs?

---

> ### Author Response · Authors · 2025-11-24
>
> We thank the reviewer for their detailed feedback and thoughtful questions. We address each concern below.
>
> >“The paper focuses on model-level vulnerability estimation, but it is unclear what the real-world application scenario of such a metric is.
>
> We thank the reviewer for their comment, and apologise that this wasn’t explained more clearly in the paper. We do not focus on auditing models against a strong adversary in the differential privacy sense, nor attempt to identify which individual records are vulnerable.
>
> MIA performance in general, and LiRA in particular, is often use by practitioners [5] as an upper bound for other types of attacks, such as reconstruction or attribute inference, under realistic attacker assumptions. As such, model vulnerability to an MIA is a valuable insight for a model developer deciding whether or not model release is safe.
>
> Our method predicts LiRA TPR at low FPR, which focuses on worst-case vulnerability across all training samples; this is in contrast with MIA AUC, which reports average risk across all records.
>
> Our main contribution is to offer an efficient estimator of this worst-case risk that does not require training reference models. While our approach does not identify which individual records are vulnerable, it does quantify the magnitude of their risk. In contrast, prior work [2,3] proposes methods to locate vulnerable samples, but, as our results show, these do not provide insight into how large the resulting risk is.
>
> Concrete application scenarios for our method include:
> - **Hyperparameter selection** - compare dozens of training configurations (learning rates, architectures, regularization) that would be prohibitively expensive with LiRA/RMIA [5]
> - **Continuous monitoring** - flag unusually vulnerable models during regular retraining
> - **Pre-deployment assessment** - quickly evaluate model privacy before sharing, without training hundreds of reference models.

---

> ### Author Response · Authors · 2025-11-24
>
> >“The technical novelty of the work is also limited. The core contribution amounts to computing the True Negative Rate (TNR) of a standard loss-based attack. This idea lacks theoretical grounding or statistical justification. The absence of analytical insights weakens the contribution, making it less suitable for a top-tier venue such as ICLR.”
>
> The reviewer is correct to point out that our method is quite simple, and applying LOSS attack itself is not novel. However, the key contribution of our work is the insight that LOSS-based statistics can reliably predict the performance of a far more expensive attack like LiRA – a connection that, to our knowledge, has not been established before and that we believe is valuable for the field. In this sense, the simplicity of the method becomes an advantage: it is extremely cheap and easy to implement, and practical for real-world model evaluation workflows. We include an extended explanation of the intuition behind our method in Appendix D.

---

> > ### Author Response · Authors · 2025-11-24
> >
> > >“The evaluation is restricted to LiRA and RMIA, both of which rely on output-distribution differences. However, many MIAs operate under different assumptions: Reference-calibrated or label-only attacks (e.g., He et al., 2024; Ye et al., 2022) rely on label confidence or query perturbations, not continuous loss values. It remains unclear whether the proposed estimator generalizes to those attack families.”
> >
> > We agree with the reviewer that exploring vulnerability to other MIAs is an important avenue for future work, since MIA vulnerability and threat models vary across setup.
> >
> > In this paper, we focus specifically on vulnerability to LiRA, which is established as the state-of-the-art and typically matches or exceeds the performance of alternative reference-model–based attacks when given the same number of reference models [2,6].
> >
> > More specifically, regarding these two attacks:
> >
> > Ye et al., 2022 [2] introduce Attack R, a loss-based reference-calibrated attack similar to LiRA but weaker under comparable assumptions and attacker knowledge [1, 6]. We therefore do not aim to predict vulnerability to Attack R (or other weaker attacks) in our main results. We agree, however, that studying how our method performs as a predictor of attacks of different strengths, including future, stronger ones, is an interesting and important direction. In Section 5.1 of our paper, we already vary LiRA’s strength by changing the number of reference models. We now additionally evaluate how well our method predicts vulnerability to Attack-R and RMIA with 64 shadow models, and find that it successfully predicts both and outperforms all baselines we consider. We include this experiment in Appendix E.
> >
> > He et al., 2024: We agree that evaluating how well our method predicts vulnerability to other families of attacks, including He et al. (2024), would be interesting future work. However, we consider these attacks to be outside the scope of this paper. Our approach relies on the connection between model loss and LiRA performance, which does not directly extend to attacks that do not use loss-based signals.

---

> > > ### Author Response · Authors · 2025-11-24
> > >
> > > >“Finally, although the authors compare their approach with metrics like LT-IQR AUC and train-test gap, these baselines are not designed for model-level vulnerability estimation. As a result, the comparison does not convincingly demonstrate the proposed method’s superiority or distinct advantages.”
> > >
> > > The task of predicting model-level vulnerability with few or no reference models is indeed a new task with no obvious direct baseline. We explain below our reasoning for the baseline we provide but are happy to consider others.
> > >
> > > We include RMIA because it is currently the most efficient high-performing attack, requiring only two reference models. While train-test gap is not a baseline per se, the connection between overfitting and vulnerability to MIAs has been studied extensively in the literature, making it a natural baseline that we considered important to include [1,3,4]. Finally, LT-IQR is a recent work proposing a method for identifying vulnerable samples “for free”. We include it to test whether such sample-level scores could be aggregated into a meaningful model-level estimate.
> > >
> > > To the best of our knowledge, there are no other metrics designed specifically for estimating model-level vulnerability at a low cost. We are happy to include additional baselines if the reviewer has specific suggestions.
> > >
> > > We hope these revisions address your concerns. Please let us know if there are any remaining issues, suggestions, or questions that -- if clarified or improved -- would contribute to raising the overall score.
> > >
> > > References
> > >
> > > [1] Carlini, Nicholas, et al. "Membership inference attacks from first principles." 2022 IEEE symposium on security and privacy (SP). IEEE, 2022.
> > >
> > > [2] Ye, Jiayuan, et al. "Enhanced membership inference attacks against machine learning models." Proceedings of the 2022 ACM SIGSAC conference on computer and communications security. 2022.
> > >
> > > [3] Yeom, Samuel, et al. "Privacy risk in machine learning: Analyzing the connection to overfitting." 2018 IEEE 31st computer security foundations symposium (CSF). IEEE, 2018.
> > >
> > > [4] Salem, Ahmed, et al. “ML-Leaks: Model and Data Independent Membership Inference Attacks and Defenses on Machine Learning Models.” NDSS Symposium. 2019.
> > >
> > > [5] Ponomareva, Natalia, et al. "How to dp-fy ml: A practical tutorial to machine learning with differential privacy." Proceedings of the 29th ACM SIGKDD Conference on Knowledge Discovery and Data Mining. 2023.
> > >
> > > [6] Zarifzadeh, S., Liu, P., & Shokri, R. (2024, July). Low-cost high-power membership inference attacks. In Proceedings of the 41st International Conference on Machine Learning (pp. 58244-58282).

---

### Official Review · Reviewer_Lyxz · 2025-10-31

**Soundness:** 4
**Presentation:** 4
**Contribution:** 4
**Rating:** 8
**Confidence:** 4

**Summary:**

The paper proposes a new method to estimate a model’s vulnerability to membership inference attacks without training any reference models. It is shown that samples most vulnerable to LiRA are the ones that were moved from the high-loss tail of the distribution to the low-loss region during training. By analyzing the loss distributions, the MIA vulnerability can be predicted. The approach outperforms SOTA methods such as RMIA in predicting LiRA’s overall success rate, while requiring no reference models.

**Strengths:**

- The proposed method is way more efficient than previous approaches.
- The method is a very good and efficient indicator to approximate vulnerability to MIAs after training a model.
- The paper was very easy to read and to follow.
- With an adaptation of the LOSS TNR to the LOSS AUC, the method can even be applied to LLMs.

**Weaknesses:**

- While LiRA and RMIA are computationally more demanding, these attacks can be used to predict membership for individual samples. The proposed method cannot predict membership for individual samples, but only estimates the vulnerability to membership inference attacks on a model level.

Misc:
- In line 82, the sentence seems to be incomplete and has "achieve" two times within the sentence.
- In line 260, "Appendix" and the closing brackets are missing.

**Questions:**

Q1: Is it somehow possible to extend this approach to allow for sample-level membership predictions?
Q2: Why use the LOSS AUC only for LLMs? Did you also try it for other "traditional" models?

**Details Of Ethics Concerns:**

There are no concerns.

---

> ### Author Response · Authors · 2025-11-24
>
> We sincerely thank the reviewer for the positive assessment of our work, particularly noting the efficiency improvements, the effectiveness of our method as a vulnerability indicator, and the clarity of presentation. We respond to the individual points and questions below.
>
> >“While LiRA and RMIA are computationally more demanding, these attacks can be used to predict membership for individual samples. The proposed method cannot predict membership for individual samples, but only estimates the vulnerability to membership inference attacks on a model level.”
>
> Our apologies that this was not made clear in the paper. We distinguish between two important goals in this area: (i) sample-level membership inference i.e which samples are most vulnerable to an attack, and (ii) estimating a model’s overall vulnerability to membership inference attacks (model-level risk i.e. TPR@low FPR which quantifies how many members can be identified with high confidence). While both are important tasks, **we view our work as complimentary to sample-level membership inference, rather than a replacement for it.** Model-level MIA vulnerability could be used by practitioners (Ponomareva et al., 2023) to answer questions like: “Is this model safe to deploy?”, “Which of these candidate architectures has lower privacy risk?” and “Does this privacy mitigation technique justify the loss in utility?”. In practice, our method can be used to estimate the magnitude of model-level vulnerability, before running a costly attack such as LiRA, helping practitioners decide whether such an attack is warranted.
>
> >“Why use the LOSS AUC only for LLMs? Did you also try it for other "traditional" models?”
>
> Thank you for this question. **We did experiment with LOSS AUC for traditional models, and these results are reported in Appendix B**. We have found LOSS TNR to be a more accurate estimator for traditional models, i.e., those where the member and non-member distributions are clearly distinct. For LLMs, where the two distributions are almost fully overlapping, we found LOSS AUC to be more successful. We report the best estimators in the main body of our paper, and  include results for other metrics including AUC in Appendix B.
>
> References
>
> Ponomareva, N., Hazimeh, H., Kurakin, A., Xu, Z., Denison, C., McMahan, H. B., ... & Thakurta, A. G. (2023). How to dp-fy ml: A practical guide to machine learning with differential privacy. Journal of Artificial Intelligence Research, 77, 1113-1201.

---

### Official Review · Reviewer_e8BK · 2025-11-01

**Soundness:** 3
**Presentation:** 3
**Contribution:** 3
**Rating:** 4
**Confidence:** 3

**Summary:**

The paper propses a low cost method to identify points that will be vulnerable to privacy leakage. The method works by tracking the loss of the points in the training set and comparing to the test set. The author further conduct experiments on LLMs.

**Strengths:**

The paper makes solid contributions. It substantially reduces the computational cost of estimating membership inference vulnerability by removing the need for reference models. It also establishes a strong empirical relationship between true MIA performance and its proposed proxy (the LOSS TNR metric), demonstrating that simple loss-based statistics can reliably estimate privacy risk. Finally, it conducts extensive experiments across diverse architectures and datasets, reinforcing the robustness and generalizability of its findings.

**Weaknesses:**

My biggest concern is with the limited scale of the image experiments. The tails tend to disappear when the generalization gap is low. For example, finetuning a large transformer models (like ViT) on CIFAR datasets. I think this represents an important case for the authors to consider.

**Questions:**

above

---

> ### Author Response · Authors · 2025-11-24
>
> We thank the reviewer for their thoughtful and constructive feedback, and for recognizing the contributions of our work, particularly the computational efficiency gains, and the strong empirical relationship we establish between LOSS TNR and LiRA TPR. We appreciate the suggestion to run these new experiments, which we feel has meaningfully improved the paper.
>
> > My biggest concern is with the limited scale of the image experiments.
>
> We agree that the separation between member and non-member loss distributions varies across setups, and that understanding this variation is important to understand the applicability of our method. **In response, we now evaluate our method on models spanning a broad range of generalization gaps**. For each target model, we treat intermediate checkpoints recorded every 10 epochs as additional target models. For each of these, we instantiate LiRA and compute the corresponding LOSS TNR. These experiments confirm that the linear relationship between LiRA TPR and LOSS TNR holds across the full range of generalization gaps, and we report the results and details in Section 5.3 of the revised paper.
>
> **We also now include results with a ViT model finetuned on CIFAR-10**, and show that the established relationship between LOSS TNR and LiRA TPR still holds.
>
> We hope these revisions address your concerns. Please let us know if there are any remaining issues, suggestions, or questions that -- if clarified or improved -- would contribute to raising the overall score.

---

### Author Response · Authors · 2025-12-03

We thank all reviewers for their thoughtful feedback. We have revised our paper to address the concerns raised and believe these improvements have significantly strengthened the paper. Below we summarize our key contributions and the revisions made during the rebuttal period.

**Key Contributions:**

We propose a simple, computationally efficient method to estimate model vulnerability, i.e. LiRA TPR at low FPR [1], without training any reference models. Our key observation is that samples vulnerable to LiRA are those whose loss values move from the tail (high loss) to the head (low loss) of the loss distribution during training, and that the fraction of such samples provides a good approximation of LiRA performance. We show that LOSS TNR, a simple and efficient metric, accurately captures this fraction and successfully predicts LiRA performance. This enables developers to assess whether models are safe to deploy, or when to apply mitigations. We demonstrate that this relationship holds across diverse architectures, datasets, and training configurations, from image classifiers to LLMs. This provides practitioners with a practical tool for privacy risk assessment, particularly in scenarios where reference-model attacks are computationally prohibitive, such as comparing multiple training configurations, monitoring continuous retraining, or evaluating models before deployment.

**In response to reviewer feedback, we made the following changes:**

- **Expanded experimental validation:** We now evaluate our method across models with a wide range of generalization gaps by using intermediate checkpoints of the target models as additional targets, and show that its performance is robust to the generalization gap. We further include a Vision Transformer fine-tuned on CIFAR-10 as an additional target, increasing the diversity of architectures.
- **Broader attack coverage:** We show that our method accurately predicts vulnerability to other loss-based attacks, specifically Attack-R [2] and RMIA [4] with 64 shadow models.
- **Threshold sensitivity analysis** We perform a comprehensive threshold sensitivity analysis, showing our method to be robust to the choice of LOSS attack threshold.
- **Theoretical connection:** We provide a detailed explanation of the connection between LOSS attack-based statistics and LiRA performance.
- **Clarified Practical Applications**: We articulate concrete use cases (hyperparameter selection, continuous monitoring, pre-deployment assessment [3]) and distinguish between sample-level and model-level vulnerability estimation. We emphasize that our method estimates worst-case privacy risk without requiring hundreds of reference models.
- **Improved Presentation**: Made all figures and tables self-contained, corrected notation errors, and improved clarity throughout.

Despite MIAs being considered the gold standard for assessing privacy risk, the substantial cost of training reference models limits their practicality in many model development workflows. We introduce a simple, computationally efficient method to estimate a model’s vulnerability to LiRA without training reference models, providing a practical tool for assessing privacy risks early in model development. Beyond providing a reliable, low-cost predictor of vulnerability, our work advances the field by showing that training dynamics can systematically capture privacy weaknesses at the model level, extending and complementing prior work that focused on sample-level vulnerability and individual training trajectories [5,6].

[1] Carlini, Nicholas, et al. "Membership inference attacks from first principles." 2022 IEEE symposium on security and privacy (SP). IEEE, 2022.

[2] Ye, Jiayuan, et al. "Enhanced membership inference attacks against machine learning models." Proceedings of the 2022 ACM SIGSAC conference on computer and communications security. 2022.

[3] Ponomareva, Natalia, et al. "How to dp-fy ml: A practical tutorial to machine learning with differential privacy." Proceedings of the 29th ACM SIGKDD Conference on Knowledge Discovery and Data Mining. 2023.

[4] Zarifzadeh, S., Liu, P., & Shokri, R. (2024, July). Low-cost high-power membership inference attacks. In Proceedings of the 41st International Conference on Machine Learning (pp. 58244-58282). (edited)

[5] Pollock, Joseph, et al. "Free {Record-Level} Privacy Risk Evaluation Through {Artifact-Based} Methods." 34th USENIX Security Symposium (USENIX Security 25). 2025.

[6] Leemann, Tobias, Bardh Prenkaj, and Gjergji Kasneci. "Is My Data Safe? Predicting Instance-Level Membership Inference Success for White-box and Black-box Attacks." ICML 2024 Next Generation of AI Safety Workshop. 2024.

---

### Meta-Review · Area_Chair_weG7 · 2026-01-07

**Summary:**

After carefully checking the paper, the reviews, the rebuttal, and the author-reviewer discussions, I think the weak points outweight the strong points. The reviewers were particularly concerned about the experimental reliability, the novelty of the work, and the validity of the evaluation metrics. I also believe that the novelty of the paper still needs to be strengthened. The weaknesses are not likely to be fixed in the camera-ready version. Thus, I recommend rejecting this paper.

**Reviewer Concerns:**

I think author addressed the experiment problem but didn't address the novelty problem. The score remains unchanged. I have carefully read the rebuttal. The rebuttal does not address any important concern raised by the reviewers.

**Reviewer Scores:**

The score remains unchanged. I have carefully read the rebuttal. The rebuttal does not address any important concern raised by the reviewers.

---

### Decision · Program_Chairs · 2026-01-26

Reject